# Long-Term Outcomes in Children with Congenital Toxoplasmosis—A Systematic Review

**DOI:** 10.3390/pathogens11101187

**Published:** 2022-10-15

**Authors:** Justus G. Garweg, François Kieffer, Laurent Mandelbrot, François Peyron, Martine Wallon

**Affiliations:** 1Swiss Eye Institute, Rotkreuz, and Uveitis Clinic, Berner Augenklinik, Zieglerstrasse 29, 3007 Bern, Switzerland; 2Department of Ophthalmology, Inselspital, University Hospital, 3010 Bern, Switzerland; 3Assistance Publique-Hôpitaux de Paris, Hôpital Armand Trousseau, Service de Néonatologie, 75012 Paris, France; 4Assistance Publique-Hôpitaux de Paris, Hôpital Louis-Mourier Service de Gynécologie-Obstétrique, 178 rue des Renouillers, 92700 Colombes, France; 5Inserm IAME-U1137, 75000 Paris, France; 6Hospices Civils de Lyon, Hôpital de la Croix Rousse, Department of Parasitology and Medical Mycology, 69004 Lyon, France; 7Walking Team, Centre for Research in Neuroscience in Lyon, 69500 Bron, France

**Keywords:** congenital toxoplasmosis, long-term outcomes, follow-up, treatment outcome, retinochoroiditis

## Abstract

Even in the absence of manifestations at birth, children with congenital toxoplasmosis (CT) may develop serious long-term sequelae later in life. This systematic review aims to present the current state of knowledge to base an informed decision on how to optimally manage these pregnancies and children. For this, a systematic literature search was performed on 28 July 2022 in PubMed, CENTRAL, ClinicalTrials.gov, Google Scholar and Scopus to identify all prospective and retrospective studies on congenital toxoplasmosis and its long-term outcomes that were evaluated by the authors. We included 31 research papers from several countries. Virulent parasite strains, low socioeconomic status and any delay of treatment seem to contribute to a worse outcome, whereas an early diagnosis of CT as a consequence of prenatal screening may be beneficial. The rate of ocular lesions in treated children increases over time to 30% in European and over 70% in South American children and can be considerably reduced by early treatment in the first year of life. After treatment, new neurological manifestations are not reported, while ocular recurrences are observed in more than 50% of patients, with a mild to moderate impact on quality of life in European cohorts when compared to a significantly reduced quality of life in the more severely affected South American children. Though CT is rare and less severe in Europe when compared with South America, antenatal screening is the only effective way to diagnose and treat affected individuals at the earliest possible time in order to reduce the burden of disease and achieve satisfying outcomes.

## 1. Introduction

Even in the absence of manifestations at birth, children with congenital toxoplasmosis (CT) may develop serious long-term sequelae, including hydrocephalus, seizures, and cognitive, hearing and visual impairments, with retinochoroiditis being the most frequent manifestation of CT. The risk for its development later in life has been linked to the initiation and timing of antibiotic treatment after birth. In a South American population, starting antiparasitic therapy as early as possible compared to a delay until the fourth month of life or later has been reported to reduce the risk of ocular lesions within the first 5 years of life from 78% to 33% [1]. While a relatively high incidence of new retinochoroidal lesions during the follow-up period indicates the importance of long-term follow-up for patients with CT in South America, its usefulness in children with asymptomatic infection at birth has been debated in North American and European countries. The underlying systematic review aims to summarize the current state of knowledge on congenital toxoplasmosis and its long-term outcomes to base an informed decision on how to optimally manage these children in the long term.

## 2. Results

### Information on Included Studies

Our systematic literature search identified 195 potential references. After the removal of duplicates, 31 articles matching our inclusion criteria were identified (see Figure 1).

The main characteristics, as well as the extracted information of the included studies, are summarized in Table 1 and Table 2. In these tables, we attempted to separate papers referring to overlapping or the same cohorts but published addressing different study questions by dashed lines, while independent cohorts are separated by continuous lines. Since overlaps between reports from similar or same cohorts could not be separated with sufficient certainty, we presented any relevant information per paper instead of per cohort. For the same reason, it was not possible to calculate means or compare cohorts and their treatments in the, therefore, purely descriptive Table 1 and Table 2.

## 3. Discussion

The risk of vertical transmission and the parasite load in the fetal tissue seems to be widely hosted and species-specific. Not surprisingly, findings from animal models are not directly transferable to the human situation, while they may provide clues to specific aspects of vertically transmitted toxoplasmosis [49]. Similar to the human situation, the risk and severity of the congenital infection are linked to the gestational age of maternal infection in mouse models of CT, and a lower number of Toxoplasma tissue cysts have been reported in the healthy retina than in the brain [3,50,51].

The virulence of T. gondii and the severity of clinical manifestations have been linked to the clonal lineage of the parasite. Three archetypical lineages have been identified, decreasing in their virulence from type I to III, while atypical strains predominant in South America have been linked to the most severe manifestations of human disease [52]. The strain type may thus at least partially explain the risk of vertical transmission and the severity of congenital manifestations in children with CT and a more severe clinical disease in South American patients who have a five-fold higher risk of developing ocular lesions which are larger and more likely centrally located [20,53,54,55,56].

Until recently, retinal cysts were not generally considered a problem in immunocompetent individuals, except in the situation of CT. Meanwhile, recurrences of toxoplasmosis have been reported in both congenital and acquired diseases, so recurrences cannot be deemed a specific marker of congenitally acquired infection [2]. Most information pertaining to the long-term outcomes of CT from the last two decades tracks cohorts of children either treated before and after or at least after birth for a minimum of 3 months, frequently 12 or more months. Information on the evolution of a disease in untreated cases merely refers to historical cases [30]. In earlier cohorts (i.e., before the mid-1990s, diagnostic and therapeutic attitudes were less precise and differed markedly from today’s practice. Routine diagnostic confirmation in serologically not unequivocally confirmed CT included mouse inoculation, resulting in a delay of up to 4 weeks for the definitive diagnosis and start with therapy [57,58], resulting in a relevant number of late and undiagnosed cases. Earlier diagnosis and fetal treatment have led to better outcomes not only in South American children [1] but also in European children with CT born after 1995, as has been reported [33,59]. In some European countries, treatment is initiated after confirmation of maternal seroconversion during pregnancy, with the objectives of reducing the risk of transmission to the fetus and the severity of the disease after vertical transmission [3]. Moreover, since no treatment is effective against bradyzoites, therapeutic effects can only affect the parasite load and, along with it, the risk and severity of organ manifestations as well as recurrence rates [22]. As late manifestations are known to arise, even in treated cases, long-term clinical monitoring is a prerequisite for detecting potential treatment effects [56].

A beneficial treatment effect is strongly supported by evidence from animal models [60,61,62]. This has rendered a placebo-controlled randomized comparison of treatment effects ethically questionable. The favourable outcomes reported compared to untreated cases imply that any comparative treatment study in humans will need a follow-up of many years before detecting differences in treatment effects [63]. Because of decreasing prevalence and the lack of randomized controlled studies with sufficient evidence in favour of early diagnosis and treatment, several public health authorities have concluded that screening in order to achieve an early diagnosis and treatment during pregnancy is not justified (i.e., Switzerland, Denmark, England and Norway) [64,65]. Central arguments are the cost-benefit ratio [7,19,66] and the induction of anxiety as a consequence of an early diagnosis in the absence of curative treatment options [57]. In moderate prevalence areas with predominantly type II strains and in high prevalence areas with atypical virulent strains, prenatal screening has been reported cost-effective as compared to neonatal screening or to the absence of screening [67,68,69]. A recent prospective multicenter randomized trial compared prenatal therapy with pyrimethamine + sulfadiazine (PY/SA) vs. spiramycin to reduce placental transmission of toxoplasmosis and showed a trend toward lower transmission with PY/SA, although it did not reach statistical significance, possibly because of a lack of statistical power [70]. There were also no fetal cerebral lesions in the PY/SA group vs. 8.6% in the spiramycin group (P = 0.01). Several observational studies reported the benefit of prenatal prophylactic treatment with PY/SA, while others failed to show any benefit [59,71,72]. After reviewing the available epidemiological evidence through 2021, the French College of Gynecology and Obstetrics recently recommended continuing the prenatal screening and treatment program, including therapy with spiramycin before gestational week 14 and PY/SA in combination with folinic acid from week 14 onwards in case of confirmed maternal infection in order to reduce the risk of vertical transmission and the risk of neurologic sequelae [73,74].

Beyond a cohort of untreated children born before 2010 in the United States, 84% presented with severe organ manifestations within the first half year of life [31,33,75], which is not better than what was reported from historical studies [76,77]. Thus, CT, if diagnosed late and left untreated, remains a deleterious disease [34,78]. In the same period, only 15–23% of treated children with confirmed CT were reported to have developed organ manifestations until the end of their first year of life in European countries, where screening used to be mandatory [18]. This provides indirect evidence of the impact of treatment in areas with less virulent parasite strains. It also argues in favour of screening in order to reduce the risk of mother-to-child transmission and, in the case of fetal infection, the parasite load in order to reduce the risk of intracranial and ocular damage to the child [31,56,73,79]. Because of the absence of screening, the gestational age at which maternal infection is acquired is not known in the United States. The lack of early diagnosis and treatment of the mother during gestation in the United States is presumably a major (although not the only) contributor to the worse outcomes at birth in this country in comparison to Europe [56]. The relevance of the strain type in a given case may play an as-yet underestimated role [53,80,81]. Atypical strain I-derived lineages have been found to lead to more severe organ manifestations [75,82], whereas the overwhelming majority of strains identified in Europe belong to low-virulent type II lineages [83,84,85]. On the other hand, type II strains are more likely associated with recurrences of ocular toxoplasmosis, at least in European patients [86].

In South America, mortality in new-born children with CT is not unusual, and 35% of children present with severe neurological disease, including hydrocephalus, microcephaly and mental retardation; 80% of children harbour ocular lesions, and up to 40% of children may present with hearing loss [20,44,81,87]. Parasite virulence may be a driving force for these extremely poor outcomes [81], but the socioeconomic situation and social and eating behaviour may also contribute to the severity of the disease [65,72,88]. Presumably, a quantification of treatment effects will be easier in severely affected cohorts, and indeed, few recent studies have found support for worse outcomes in insufficiently treated instances of severe fetal toxoplasmosis [1,69,70,89].

Taken together, the mentioned points indicate that virulent parasite strains, low socioeconomic standards and the absence or delay of treatment contribute to a worse outcome, whereas there exists some evidence that early diagnosis of CT as a consequence of prenatal screening is beneficial [70]. Arguments in favour of screening and pre- and postnatal treatment of affected children are overwhelming [59].

### 3.1. Eye Disease, Recurrences and Impact of Treatment

From the clinical picture of an ocular lesion, it is virtually impossible to differentiate a congenital from an acquired lesion [90]. Specific IgM is not usually found in both instances since in acquired as well as CT, and ocular precipitation usually arises significantly after primary parasitaemia [2,91]. Therefore, in the ophthalmological literature, a differentiation between congenital and acquired cases has to be rendered impossible [90]. Any information pertaining to the management and long-term outcome of ocular lesions in CT herein will have to be derived from cases in which the congenital route of acquisition has been confirmed in advance.

Unbiased data from untreated cohorts are not available for European patients [76,92], and reports from American referral centres are alarming [31] though exposed to a significant selection bias with referral only of the more severe cases. In treated instances, however, the severity of disease in general and specifically of ocular lesions may be less [35]. The rate of ocular lesions in treated children depends on the length of follow-up, frequently before the age of 5 years [8,15]. Nevertheless, first lesions may arise more than 12 years after birth [17,93]. In European patients, lesions are present in 17% of children after 3 years [21] and in 24% of children after a follow-up of 6 years [9]. Expanding the follow-up to more than 10 years, ocular lesions were found in 29.8% of treated children, which were unilateral in 69.0% and did not cause vision loss in 80.6%. Bilateral visual impairment thus seems rare in European cohorts, and two-thirds present with normal vision in either eye [9,15,21].

During long-term follow-up, 33.8% of children with CT will develop recurrences or new ocular lesions up to 12 years after the appearance of the first lesion [93]. Interestingly, only every sixth lesion (17%) seems to be active at diagnosis. In treated European children, secondary ocular pathologies such as squinting, cataracts and microphthalmia were reported in 19%, which is below expectations from other series [34,94]. These are generally linked to a more severe affection and to poor visual function. Unilateral visual impairment was reported in 24% of treated children. Beyond secondary pathologies, squinting (strabismus) was the most prevalent (16%). The overall functional prognosis of CT in Europe is better than would be expected on the basis of literature findings, between less than 2% and 9% suffering bilateral visual impairment [4,21]. The consequences of CT are rarely severe in European children treated in utero and until the end of their first year of life. Nevertheless, annual postnatal monitoring is justified, owing to the persisting risk of ocular affection [8,10,33]. Indeed, regular follow-up is estimated by the wide majority of affected individuals (98%), and in most instances (92%), it was reported as reassuring, although 11% of children, namely patients with low visual acuity and low visual function scores, found the follow-ups frightening [95].

In North American patients, macular involvement was reported in 54% and bilateral involvement in 41% of instances with confirmed congenital ocular toxoplasmosis [Mets, 1996 #38]. Some evidence exists that the severity of clinical manifestations is related to delayed onset and shorter duration of therapy [1,34]. In South America, the burden of disease [44,47,96], namely the incidence and severity of ocular manifestations and their impact on visual function, are much more severe, with 60-80% of affected individuals presenting with ocular lesions [20,39,87,94]. Recurrences are observed in more than 50% of patients and are more frequent than in Europe [33]. The recurrence rate, in contrast, does not differ [97]. In Brazilian patients, ocular involvement is observed in up to 70% already short after birth, the majority located in the central retina, and in two-thirds (65.9%) of patients, the disease is already bilateral. Secondary eye complications are uncovered in 50% of patients, with cataracts, microphthalmia and strabismus being the most prevalent [39]. In a recent study from Brazil, a prevalence of CT of 1.2 cases/1000 live births were reported [98]. In another Brazilian cohort, retinochoroiditis was reported in 71.4% of patients. As mentioned before, new retinochoroidal lesions developed after the first year of life in 77.8% of patients who were treated starting 4 months after birth compared to 33.3% of patients if treated before 2 months of life (relative risk = 0.42, 95% confidence interval: 0.25–0.72, *p* = 0.01). Evidently, early treatment initiation strongly affects the degree and severity of organ damage. Interestingly, two peak incidences of new retinochoroidal lesions were found in this cohort between 4 and 5 years and between 9 and 14 years, with the latter only among girls [1].

Macular lesions and bilateral affection are more frequently reported in congenital disease compared to postnatal disease in European cohorts. Moreover, the congenital disease seems to be a risk factor for vision loss in ocular toxoplasmosis [17,99]. In ocular toxoplasmosis cohorts not differentiating congenitally attracted and acquired cases, visual function is generally good. A functional impact of ocular lesions is found in the visual field in 94% of eyes, whereas visual acuity is normal in 59% of eyes and nearly normal (>20/40) in an additional 13.5% of eyes, showing perimetric findings to be more sensitive in quantifying chorioretinal damage [100,101]. Moderate to severe functional impairment may be registered in 65.2% of patients for visual field and in 27.5% of patients for visual acuity [100]. Bilateral macular affection is associated with reduced cognitive functional scores. Although children with bilateral visual impairment seem to compensate with higher verbal skills, their verbal scores are still less than those of children with normal vision, indicating that their intellectual performance is generally reduced [102]. In European cohorts, when diagnosed and treated early, CT has little effect on the quality of life and visual function of the affected individuals [11]. This must not be generalized to other parts of the world. In South America, in contrast, quality of life seems to be much more compromised in less early diagnosed, less well-treated and more severely manifested instances, as is the rule in South America [55,103,104].

### 3.2. Postnatal Neurological Evolution

While much has been reported about neurological manifestations at birth and the long-term evolution of the ocular disease, literature regarding the postnatal evolution of the neurologic disease is scarce. This may at least partially be attributed to the technical problem of imaging in cerebral pathologies, which is far more complex than that for ocular disease, frequently requiring the use of sedatives. As a consequence, routine clinical imaging has not systematically been applied as long as new neurological findings or changes are observed.

Generally, an incidence of neurologic findings in treated European children with CT of around 12% must be expected compared to almost 60% for ocular lesions [11]. In a series of 24 North American children with untreated and asymptomatic CT, in contrast, Wilson, Remington and colleagues reported severe permanent neurological sequelae after developing eye disease in three instances and mental retardation with an IQ between 36 and 62 in four instances, while six more patients developed mild cognitive impairment over a follow-up of 5.5 years, adding up to a significant neurological impact in 40% in this cohort [78]. In 1985, Kaiser reported about 10 European children requiring a shunt for the treatment of hydrocephalus, nine of whom also presented with ocular manifestations. While one child died, two-thirds of the children had a reasonable outcome. He found neonatal hydrocephalus, extensive intracerebral calcification and severe ocular involvement as poor prognostic indicators [105]. Ventriculoperitoneal shunt placement in patients with CT and even severe hydrocephalus can lead to favourable clinical and cognitive outcomes if performed early [106]. Both former series refer to children that were only treated after manifesting symptomatic disease. In these patients, severe neurological disease was linked to ocular lesions in the majority of instances [78,105]. This association was also reported in a large French cohort in which extraocular CT lesions at baseline were associated with a higher risk of retinochoroiditis [93]. In contrast to earlier reports, McAuley and colleagues documented in their series of 44 children normal developmental, neurological and ophthalmologic findings at the follow-up of many of the treated children despite severe primary manifestations, including substantial systemic disease, hydrocephalus, microcephalus, multiple intracranial calcifications, and extensive macular destruction detected at birth. The authors attributed this success to the 1-year duration of antiparasitic therapy given to these children after birth [34]. There is additional evidence from prenatal series that early antiparasitic therapy is associated with a decreased risk of neurological symptoms in children with congenital toxoplasmosis [15,19,106].

In a North American series of 52 children with CT diagnosed at birth, 40% presented with ocular and/or neurologic manifestations. Only one child developed a neurologic deficit resulting from a congenitally present lesion [37]. In a prospective longitudinal North American study including 36 patients with CT, neurologic and developmental outcomes were significantly better in children treated for the first year of their life compared to that reported in children treated for 1 month or less. While their cognitive performance remained stable over time, it was lower in treated children when compared to that of their uninfected siblings. Thus, a robust benefit of treatment is likely encountered in neurologic and ocular disease if the diagnosis and treatment of pregnant women with acute gestational toxoplasmosis and their offspring with CT are affected early [102]. Intracranial calcifications diminished or resolved in 75% of 40 new-borns of the same cohort treated in the first year of life and remained stable in the remaining compared to unchanged or worsening radiological findings in children treated for 1 month or less [107]. Another study from Chicago compared the outcomes of 120 children with CT treated for the first year of their life to a historical group with untreated CT and found that treatment resulted in normal cognitive, neurologic, and auditory outcomes for all patients. If moderate to severe neurologic findings were present at birth, normal neurologic and cognitive development was achieved in >72% under treatment for the first year of life, which is remarkably better than outcomes reported for children treated for 1 month or less [35]. In line, cerebral hyperechogenic lesions without ventriculomegaly found on fetal ultrasound did not predict cognitive impairment in the majority of treated instances, although the risk of ocular lesions was high [106].

In a Brazilian cohort of 43 children with CT, a subclinical disease was present in 88% at birth. Nevertheless, half of these children developed neurologic manifestations over time, while 77% of these children presented with pathological findings on brain computed tomography. Despite treatment during the first year of life, more than half of these children developed neurologic sequelae during the follow-up period, namely delayed neuropsychomotor development, and 95% of children presented with or developed new ocular lesions. Interestingly, in this series, the presence of cerebral calcifications was not linked to a higher incidence of neurologic sequelae [108]. This confirms what has been reported for ocular disease—despite postnatal treatment for 1 year, more severe neurological disease is to be expected in a South American population with CT.

To summarize, roughly 10–15% of children with CT in a North American or European cohort will develop cerebral manifestations, with a good prognosis in the vast majority of treated instances, compared to up to 77% in their South American counterparts, 50% of whom will develop neurologic sequelae. When brain imaging is normal, the long-term neurological evolution is favourable in European and North American children [37,109]. When brain imaging shows only hyperechogenic lesions, the evolution is even in Brazilian series usually good [108,110]. In the case of hydrocephalus not requiring a shunt, the intellectual prognosis is limited, while the motor prognosis remains good. When a ventriculo-peritoneal shunt had to be inserted, the intellectual sequelae are important and more heterogeneous for motor skills [106].

## 4. Materials and Methods

A systematic literature search was last updated on 28 July 2022 in the NCBI/PubMed database from the National Institutes of Health, USA (https://www.ncbi.nlm.nih.gov/pubmed, accessed on 28 July 2022), the Cochrane Central Register of Controlled Trials (https://www.cochranelibrary.com/central/about-central, accessed on 28 July 2022), ClinicalTrials.gov, Google Scholar and the Scopus database (https://www.scopus.com/home.uri, accessed on 28 July 2022) to identify pro- and retrospective studies retrieved by the key terms congenital toxoplasmosis, long-term outcome, consequence, sequelae, outcome, effect and impact in accordance to the PRISMA (Preferred Reporting Items for Systematic Reviews and Meta-Analyses) guidelines. Further, reference lists of published meta-analyses and reviews have been screened for suitable original articles. Research articles were included if they provided data on human congenital toxoplasmosis and related long-term outcomes and if they were written in English, French or German. Papers were not included for the following reasons: reviews reporting and meta-analyses summarizing elsewhere reported patients and observations; editorial notes without reporting new observations; papers focusing on diagnosis and treatment of the pregnant woman and pregnancy outcomes at birth without later follow-up of the infected children; papers reporting on children diagnosed beyond one year of age (because of uncertainties with the time of disease acquisition).

Automatic and manual checks for duplicates have been conducted. Two independent researchers screened all references for inclusion criteria. In case of discrepancies, those were resolved by discussion. All resulting manuscripts were evaluated by the authors. Cross references identified during a manual search of references from the retrieved articles were also included if they provided additional information on congenital toxoplasmosis and its long-term outcomes. One properly trained author extracted the data from all suitable references in a two-step process: first, all necessary information based on a coding sheet draft was entered. Additional information was then added to the coding scheme. In the second step, missing data were specifically searched in the research papers. All data entries have been confirmed by the first author.

The following information has been extracted per paper (as far as available): general information such as identification (first author and publication year), dataset (in case data belong to a repeatedly used or shared dataset), number of patients included (patients with congenital toxoplasmosis or pregnant woman with transmission to the fetus), demographic information (age, gender, country, time of data collection, follow up duration), diagnosis during pregnancy or first year of life, pre- and postnatal treatment, eye-related manifestations of disease, other organ-related manifestation of disease (Table 1). We extracted total numbers (N) and frequencies from the research papers.

To avoid any kind of regional bias, we included data from all over the world. Further, we included data from countries using prenatal screening as well as from countries that do not. Last, to enable a broad picture, we included literature published between 1974 and 2022 in peer-reviewed journals. By including only peer-reviewed journals, the datasets ensure an appropriate level of methodological robustness. In general, an inherent bias cannot be avoided based on multiple variables with impact on outcomes, i.e., time of diagnosis (ante- or postnatal), treatment initiation and duration of therapy, as well as choice of antibiotic combination in this specific disease group.

To facilitate a comprehensive overview, the extracted data have been sorted and grouped by country and region. In case of missing information, related publications have been consulted.

## 5. Conclusions

In conclusion, although CT has become less frequently diagnosed in European countries, producing less harm than reported in older studies, screening is the only effective way to diagnose and treat affected individuals at the earliest possible time and to achieve the reported excellent outcomes. The burden of disease is more severe in South American countries, so tight and regular prenatal screening and the earliest possible fetal and new-born therapy are strongly recommended. Moreover, affected individuals deserve to obtain regular clinical and ophthalmological controls until they are able to report visual symptoms, minimally until the age of 8–10 years of life, in order to prevent unnecessary damage to their visual function, prevent unnecessary handicaps, and minimize the burden of disease.

## Figures and Tables

**Figure 1 pathogens-11-01187-f001:**
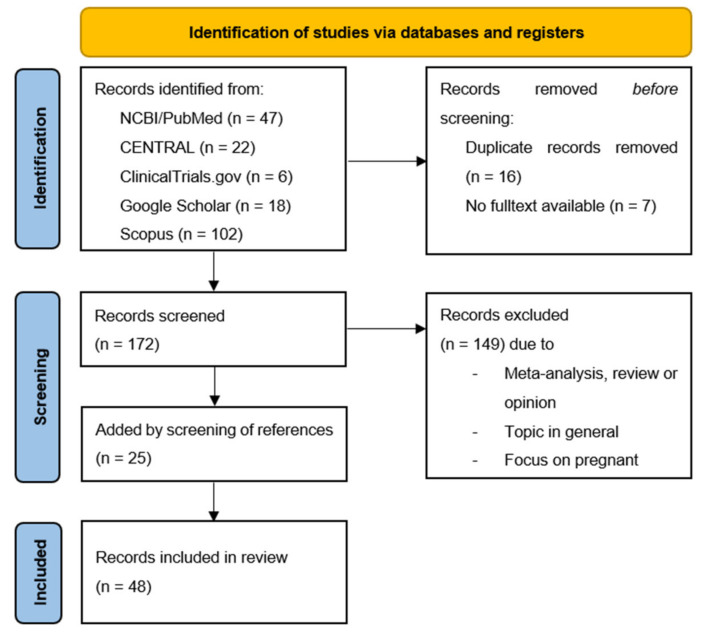
**PRISMA flow diagram.** From: Page, M.J.; McKenzie, J.E.; Bossuyt, P.M.; Boutron, I. Hoffmann, T.C., Mulrow, C.D. et al. The PRISMA 2020 statement: an updated guideline for reporting systematic reviews. *BMJ* 2021, 372, n71. doi: 10.1136/bmj.n71.

**Table 1 pathogens-11-01187-t001:** Characteristics of included studies part 1.

Reference	N	Observation Period	FU Duration	Time of Diagnosis	Extraocular Organ Manifestation within 1 Year	Eye-Related Manifestation within 1 Year	Extraocular Organ Manifestation by End of FU	Eye-Related Manifestation by End of FU
First author year, # reference nr.	Number of patients (of n pregnancies)	In years	In years	0 = during pregnancy, 1 = during first year of life (n per N)	n per N	n per N	n per N	n per N
Delair, 2008 # [2]	62	1994–2005	n. r.	0 (49 of 62); 1 (3 of 62)	n. r.	n. r.	n. r.	62 of 62
Wallon, 2013 # [3]	485 (2048)	1987–2008	3	n. r.	n. r.	n. r.	5 of 207 hydrocephaly, 22 of 207 intracranial calcifications	32 of 207 retinochoroiditis
Kodjikian, 2006 # [4]	430	1975–2001	12	n. r.	n. r.	n. r.	41 of 430 neuropathological condition (hydrocephalus, cerebral calcification, convulsion, paresis, epilepsy), 1 of 430 atrophy of the cortex	130 of 430 retinochoroiditis, 21 of 430 strabismus, 7 of 430 microphthalmia, 4 of 430 cataract
Gilbert, 2001 # [5]	181 (554)	1987–1995	n. r.	0 (141 of 181)	n. r.	n. r.	n. r.	n. r.
Gras, 2001 # [6]	181	1987–1995	3	0 (181 of 181)	n. r.	19 of 173	17 of 181 intracranial manifestation	36 of 157
Gilbert, 2001 # [7]	549	1987–1995	4.5	n. r.	n. r.	n. r.	4 hydrocephalus, 15 intracranial calcifications	33 retinochoroiditis
Garweg, 2005 # [8]	327	1988–2001	6.3	n. r.	n. r.	7 of 327	18 of 327 changes in heart, liver, spleen, hydrocephalus, microcephalus, cerebral calcification	79 of 327
Wallon, 2004 # [9]	327	1988–2001	up to 14	0 (87 of 327); 1 (163 of 327)	n. r.	38 of 317 retinochoroidal lesion	31 cranial calcifications, 6 hydrocephalus, 1 microcephalus	79 of 327 retinochoroidal lesion
Wallon, 2001 # [10]	133	1988–1993	7.9	n. r.	10 of 133	5 of 133	n. r.	34 of 133
Peyron, 2011 # [11]	102	1983–1991	22.2	n. r.	n. r.	n. r.	11 of 102 intracranial calcifications, 2 of 102 hydrocephalus, 1 seizure	60 of 102 ocular lesions, 13 of 102 reduced visual function
Kieffer, 2008 # [12]	300	1996–2002	2	0 (107 of 151 tested); 1 (139 of 300)	22 cerebral calcifications	n. r.	n. r.	36 retinochoroiditis
Kieffer, 2002 # [13]	46	1996–1998	2.3	0 (16 of 28)	3 of 46 cerebral calcification	6 of 46 chorioretinitis	n. r.	10 of 46
Desmonts, 1974 # [14]	59	n. r.	n. r.	1 (59 of 59)	n. r.	n. r.	2 of 59 death, 7 of 59 severe disease with cerebral and ocular involvement	
Berrebi, 2010 # [15]	107	1985–2005	8.9	0 (90 of 112)	n. r.	11 of 28 chorioretinitis	1 of 112 serious neurological involvement	28 of 112 chorioretinitis
Bessières, 2001 # [16]	57	1986–1996	n. r.	0 (25 of 57); 1 (32 of 57)	n. r.	n. r.	n. r.	n. r.
Faucher, 2012 # [17]	127	1995–2010	up to 12	0 (38 of 127); 1(89 of 127)	n. r.	n. r.	7 of 127 cerebral calcifications, 1 of 127 white substance modifications, 2 of 127 ventricular dilation, 1 of 127 bilateral temporal micro abscesses, 2 of 127 language development disorder	24 of 127 ocular lesions
Villena, 1998 # [18]	78	1980–1997	n. r.	0 (15 of 78); 1 (63 of 78)	7 of 78 cerebral involvement, 3 of 78 hydrocephalus, 6 of 78 intracranial calcifications	9 of 78 ocular involvement, 10 of 78 chorioretinitis, 2 of 78 microphthalmia	1 of 78 mild epileptic fits, 1 of 78 psychomotor retardation	18 of 78 chorioretinitis, 4 of 78 unilateral blindness
Cortina-Borja, 2010 # [19]	221	n. r.	4	n. r.	7 intracranial lesions, 2 lymphadenopathy/hepatosplenomegaly	5	3 deaths, 5 microcephaly, seizures, shunt required, 5 cerebral palsy or abnormal neurological development	5 ocular microphthalmia, visual opacities, cataract, strabismus, 1 blindness
**Cortina-Borja, 2010 #** [19]	**72**	**n. r.**	**4**	**n. r.**	**4 intracranial lesions, 3 lymphadenopathy/hepatosplenomegaly**	**4**	**1 death,** **4 microcephaly, seizures, shunt required,** **3 cerebral palsy or abnormal neurological development**	**5 ocular microphthalmia, visual opacities, cataract, strabismus,** **2 blindness**
Gilbert, 2008 # [20]	71	1992–2000	n. r.	n. r.	n. r.	n. r.	n. r.	16 of 70
Gilbert, 2008 # [20]	210	1996–1999	4.1	n. r.	n. r.	n. r.	n. r.	34 of 210
Tan, 2007 # [21]	281	n. r.	4.8	n. r.	n. r.	33 of 281	n. r.	49 of 281 retinochoroidal lesions
Foulon, 1999 # [22]	61	n. r.	1	n. r.	n. r.	n. r.	10 of 64 mild sequelae, 9 of 64 serious sequelae (intrauterine death, neurologic abnormalities, hydrocephalus, cerebral calcifications, chorioretinal scars)	
Koppe, 1974 # [23]	12	1964–1966	7	n. r.	0	1 squinting, 4 retinal scars	0	1 retinal scar
Meenken, 1995 # [24]	17	n. r.	27	1 (17 of 17)	10 hydrocephalus, 12 intracerebral calcifications	17 chorioretinitis	2 panhypopituitarism, 1 gonadal failure with dwarfism, 1 precocious puberty with dwarfism and thyroid deficiency, 1 diabetes mellitus and thyroid deficiency	12 of 12 chorioretinitis, 10 optic nerve atrophy, 10 visual acuity below 0.1, 5 cataract
Gilbert, 2001 # [7]	51	1987–1988	4.5	n. r.	n. r.	n. r.	1 intracranial calcification	3 retinochoroiditis
Gilbert, 2001 # [7]	133	1992–1995	1.25	n. r.	n. r.	n. r.	2 intracranial calcifications	3 retinochoroiditis
Gilbert, 2001 # [7]	123	1992–1996	3.2	n. r.	n. r.	n. r.	1 hydrocephalus, 2 intracranial calcifications	4 retinochoroiditis
di Carlo, 2011 # [25]	14	2003–2008	4.6	0 (7 of 50 initially tested); 1 (14 of 85)	5 of 14 periventricular calcification, 5 of 14 hydrocephaly/ventriculomegaly, 3 of 14 intraparenchymateous lesions	4 of 14 retinochoroiditis	n. r.	n. r.
Galanakis, 2007 # [26]	35	1997–2003	1.2–8.2	0 (2 of 20 tested); 1 (4 of 35)	1 of 35 seizure	n. r.	3 of 35 intracranial calcifications; 1 of 35 squinting	1 of 35 retinal hemorrhage
Logar, 2002 # [27]	11	1996–1999	n. r.	n. r.	n. r.	n. r.	n. r.	1 of 11 abnormal EEG, intracranial calcification, microcrania, spastic paresis; 1 of 11 abnormal EEG, epilepsy
Fahnehjelm, 2000 # [28]	3	1997–1998	2	1 (3 of 3)	2 of 3 intracranial calcification, 1 of 3 hydrocephalus	1 of 3 chorioretinitis	n. r.	1 of 3 chorioretinitis
Boudaouara, 2018 # [29]	35	2005–2016	5.3	0 (5 of 15); 1 (30 of 35)	2	4	n. r.	n. r.
Olariu, 2019 # [30]	25	1991–2005	n. r.	1 (25 of 25)	5 of 13 hydrocephalus, 13 of 17 cerebral calcifications	10 of 16	n. r.	n. r.
Olariu, 2011 # [31]	164	1991–2005	n. r.	1 (164 of 164)	67 of 99 hydrocephalus; 94 of 118 cerebral calcifications	119 of 129	n. r.	n. r.
Phan, 2008 # [32]	38	1981–2005	5.7	1 (0 of 38)	n. r.	n. r.	26 brain calcifications, 8 hydrocephalus, 1 mild ventricular dilatation	22 of 31
Phan, 2008 # [33]	132	1981–2005	n. r.	1 (132 of 132)	n. r.	n. r.	n. r.	34 of 108 chorioretinal lesions
McAuley, 1994 # [34]	44	1981–1991	n. r.	n. r.	6 of 44 cerebral calcifications <3, 25 of 44 cerebral calcifications >3, 17 of 44 hydrocephalus	22 of 44 peripheral retinal lesions, 29 of 44 retinal lesion threatens vision	3 afebrile seizures	3 retinal lesions
McLeod, 2006 # [35]	120	1981–2004	n. r.	n. r.	n. r.	n. r.	n. r.	18 of 58 (new eye lesions); 49 of 68 vision impaired
Mets, 1996 # [36]	76	n. r.	n. r.	n. r.	n. r.	n. r.	n. r.	56 of 76 chorioretinal lesions, 44 of 76 chorioretinal scars, 41 of 76 macular scars
Mets, 1996 # [36]	18	n. r.	n. r.	n. r.	n. r.	n. r.	n. r.	18 of 18 chorioretinal lesions, 15 of 18 chorioretinal scars, 14 of 18 macular scars
Guerina, 1994 # [37]	52	1986–1992	n. r.	n. r.	9 of 46 intracranial calcifications, 1 of 47 ventriculomegaly	2 of 48 active chorioretinitis, 7 of 48 retinal scars without activity	n. r.	3 of 39 minor scars and active chorioretinitis
Neto, 2000 # [38]	47	1995–1996/8	n. r.	n. r.	n. r.	n. r.	3 intracranial calcifications, 1 hepatosplenomegaly with lymphadenopathy	5 retinal scars
Gilbert, 2008 # [20]	30	1999–2002	3.7	n. r.	n. r.	15 of 30	n. r.	20 of 30
Melamed, 2010 # [39]	44	2000–2004	n. r.	0 (25 of 44); 1 (19 of 44)	n. r.	n. r.	n. r.	31 of 44 ocular involvement, 29 of 44 retinochoroiditis, 12 of 44 strabismus, 7 of 44 nystagmus, 6 of 44 cataract, 5 of 44 microphthalmia
Lago, 2007 # [40]	6	2002	1	0 (1 of 6); 1 (5 of 6)	1 of 6 hydrocephalus; 1 of 6 hepatosplenomegaly; 3 of 6 intracranial calcifications	3 of 6 retinochoroiditis; 1 of 6 microphthalmia	1 of 6 severe mental retardation	2 of 6 strabismus, impaired vision; 1 of 6 severe visual impairment
Lago, 2009 # [41]	4	2002–2003	n. r.	0 (0 of 4); 1 (4 of 4)	1 of 4 hydrocephalus, 3 of 4 cerebral calcifications, 1 of 4 cerebral ventricular dilatation	4 of 4 retinochoroiditis; 1 of 4 microphthalmia	n. r.	n. r.
Lago, 2021 # [1]	77	1996–2017	10	0 (27 of 77); 1(46 of 77)	n. r.	39 of 55	n. r.	55 of 77 retinochoroiditis, 2 of 77 strabismus and/or optic neuritis
Soares, 2012 # [42]	58	2002–2010	n. r.	0 (3 of 19 tested); 1 (19 of 58)	n. r.	n. r.	10 of 58 intracranial calcifications, 9 of 58 delayed psychomotor development, 9 of 58 splenomegaly, 13 hepatomegaly	20 of 58 retinochoroiditis, 17 of 58 strabismus, 6 of 58 nystagmus
Andrade, 2008 # [43]	20	2003–2004	n. r.	0 (6 of 20); 1 (14 of 20)	3 of 19 liver and spleen enlargement and/or petechias, 2 of 19 hydrocephaly, 4 of 19 sensorineural impairment, 2 of 19 conductive hearing loss	2 of 19 microphthalmia	2 of 19 neuropsychomotor development delay	n. r.
Vasconcelos-Santos 2009 # [44]	190	2006–2007	n. r.	n. r.	39 of 190 intracranial calcifications, 10 of 190 microcephaly, 12 of 190 hydrocephalus	142 of 190 retinochoroidal lesions, 8 of 190 microphthalmia	n. r.	n. r.
Higa, 2014 # [45]	4	n. r.	n. r.	0 (4 of 4)	n. r.	n. r.	1 of 4 intracranial calcification, microcephaly, mental abnormalities	1 of 4 retinochoroiditis with blindness
Carvalheiro, 2005 # [46]	5	2001	1.3	n. r.	1 of 5 hydrocephalus; 1 of 5 cerebral and cerebellar calcifications	5 of 5 chorioretinitis, 2 of 5 strabismus	n. r.	n. r.
de Melo Inagaki, 2012 # [47]	6	2009	1.7	0 (0 of 6); 1 (n. r.)	cerebral calcifications 1 of 6, hepatosplenomegaly 1 of 6	retinochoroidal scars 3 of 6	n. r.	n. r.
Gómez, 2005 # [48]	26	2000–2004	1	0 (26 of 26); 1 (26 of 26)	11 of 26	10 of 26	n. r.	n. r.

Note. N = population, n = subpopulation, FU = follow up, n. r. = not reported, written in bold = mean values of all subsamples, solid line = separation of different patient samples, dashed line = studies belong to the same overall patient pool (studies belonging to a common patient pool are also marked in light grey).

**Table 2 pathogens-11-01187-t002:** Characteristics of included studies part 2.

Reference	Geographic Origin of Cohort	Therapy during Pregnancy	Postnatal Therapy		Therapy Started within x Weeks	Duration of Treatment
First author year	(with region)	1 = Spiramycin, 2 = Pyrimethamin and Sulfadiazin, 3 = other, 4 = combination, 5 = missing	1 = Spiramycin, 2 = Pyrimethamin and Sulfadiazin, 3 = other, 4 = combination, 5 = missing		0 = no, 1 = yes (n per N)	In months (n per N)
Delair, 2008 # [2]	France (single referral center)	n. r.	n. r.		n. r.	n. r.
Wallon, 2013 # [3]	France (Lyon)	1 (1616 of 2048); 2 (285 of 2048)	n. r.		n. r.	n. r.
Kodjikian, 2006 # [4]	France (Lyon)	1; 4	2	8 weeks (430 of 430)	13 to 16	
Gilbert, 2001 # [5]	France (Lyon)	1 (416 of 554); 4 (107 of 554)	n. r.	n. r.	n. r.	
Gras, 2001 # [6]	France (Lyon)	1 (89 of 181); 2 (70 of 181)	1; 2 (177 of 181)		8 weeks (177 of 181)	at least 12
Gilbert, 2001 # [7]	France (Lyon)	1; 4; 2 (108 of 549)	4 (156 of 549)		n. r.	6–12
Garweg, 2005 # [8]	France (Lyon)	n. r.	4		n. r.	12–14
Wallon, 2004 # [9]	France (Lyon)	1 (149 of 325); 2 (19 of 325); 4 (104 of 325)	2 (325 of 325)		n. r.	appr. 12
Wallon, 2001 # [10]	France (Lyon)	1 (72 of 133); 4 (42 of 133)	4		8 weeks (129 of 133)	at least 12
Peyron, 2011 # [11]	France (Lyon)	n. r.	n. r.		n. r.	n. r.
Kieffer, 2008 # [12]	France (Paris, Lyon, Marseilles)	2 (134 of 300); 1 (112 of 300)	2		4 weeks (107 of 300); 12 weeks (139 of 300)	n. r.
Kieffer, 2002 # [13]	France	2 (16 of 46); 1 (30 of 46)	2		4 weeks (37 of 46)	12
Desmonts, 1974 # [14]	France (Paris)	1	no treatment	n. r.	0	
Berrebi, 2010 # [15]	France (Toulouse)	1 (22 of 112); 2 (90 of 112); 4 (6 of 112)	2	4 weeks	n. r.	
Bessières, 2001 # [16]	France (Toulouse)	1; 4 (overall 57 of 57)	n. r.	n. r.	n. r.	
Faucher, 2012 # [17]	France (Marseilles)	1 (52 of 127); 2 (5 of 127); 4 (45 of 127)	2 (127 of 127)	n. r.	12	
Villena, 1998 # [18]	France (Reims)	1 (66 of 78)	2; 4	n. r.	12–24	
**Cortina-Borja****, 2010****#** [19]	**France, Austria, Italy**	**1 (87 of 221); 2 (102 of 221)**	**n. r.**	**n. r.**	**n. r.**	
**Cortina-Borja****, 2010****#** [19]	**Denmark, Sweden, Poland**	**n. r.**	**n. r.**		**n. r.**	**n. r.**
Gilbert, 2008 # [20]	Europe (Poland and Scandinavia)	0	2; 4		4 weeks	12
Gilbert, 2008 # [20]	Europe (France, Italy and Austria)	1; 2; 4	2; 4		4 weeks	12
Tan, 2007 # [21]	Europe	n. r.	n. r.		n. r.	n. r.
Foulon, 1999 # [22]	Europe (Helsinki, Oslo, Brussels, Lille, Reims)	1 (39 of 46 treated); 2 (7 of 46)	most congenitally infected infants were treated postnatally during first year of life		n. r.	n. r.
Koppe, 1974 # [23]	The Netherlands	n. r.	2 (5 of 12)		n. r.	0
Meenken, 1995 # [24]	The Netherlands	n. r.	2 (9 of 17), 4 (2 of 17)		n. r.	2–10
Gilbert, 2001 # [7]	The Netherlands	3 (spiramycin and sulfadiazin)	n. r.		n. r.	n. r.
Gilbert, 2001 # [7]	Austria	1; 4	4 (34 of 133)		n. r.	6–12
Gilbert, 2001 # [7]	Denmark	n. r.	4 (26 of 123)		n. r.	6–12
di Carlo, 2011 # [25]	Italy	1; 2	4 (14 of 14)		4 weeks (14 of 14)	0
Galanakis, 2007 # [26]	Greece (Crete)	1 (31 of 35); 2 (2 of 35); 3 (2 of 35; roxithromycin)	2 (35 of 35)		n. r.	2–12
Logar, 2002 # [27]	Slovenia (Ljubljana)	2 (9 of 11)	1; 2; 4		n. r.	8–12
Fahnehjelm, 2000 # [28]	Sweden (Stockholm, Skåne)	n. r.	4		4 weeks (2 of 3); 8 weeks (1 of 3)	12
Boudaouara, 2018 # [29]	Tunisia	1 (22 of 35); 2 (5 of 35); 5(8 of 35)	2 (34 of 35)		4 weeks (34 of 35)	12 (34 of 35)
Olariu, 2019 # [30]	US	n. r.	n. r.		n. r.	n. r.
Olariu, 2011 # [31]	US, Canada	n. r.	n. r.		n. r.	n. r.
Phan, 2008 # [32]	US	n. r.	n. r.		0	0
Phan, 2008 # [33]	USA, Canada	1 (1 of 108); 2 (12 of 108); 4 (5 of 108); 3 (1 of 108 gantrisin)	2 (132)		12 weeks	12
McAuley, 1994 # [34]	USA (Chicago), Hawaii, Canada	n. r.	4 (4 of 35); 2 (31 of 35)		n. r.	12
McLeod, 2006 # [35]	USA (Chicago), Hawaii, Canada	n. r.	2		12 weeks	12
Mets, 1996 # [36]	USA (Chicago), Canada, Mexico	n. r.	2 (69 of 76), 3 (7 of 76)		4 weeks	12
Mets, 1996 # [36]	USA (Chicago), Canada, Mexico	n. r.	0		n. r.	n. r.
Guerina, 1994 # [37]	USA (Massachusetts, New Hampshire)	n. r.	2 (47 of 49); 1 (2 of 49)		n. r.	12 (45 of 49)
Neto, 2000 # [38]	Brazil (Rio Grande do Sul)	n. r.	n. r.		n. r.	n. r.
Gilbert, 2008 # [20]	Brazil (Rio Grande do Sul)	n. r.	2; 4		8 weeks	12
Melamed, 2010 # [39]	Brazil (Rio Grande do Sul)	n. r.	2; 3 (prednisone)		n. r.	n. r.
Lago, 2007 # [40]	Brazil (Rio Grande do Sul)	1 (1 of 6)	2 (3 of 6); 4 (3 of 6, prednisone, zidovudine)		4 weeks (3 of 6); 8 weeks (1 of 6); 12 weeks (2 of 6)	n. r.
Lago, 2009 # [41]	Brazil (Rio Grande do Sul)	n. r.	2 (4 of 4)		n. r.	12
Lago, 2021 # [1]	Brazil (Rio Grande do Sul)	1 (13 of 77); 2 (7 of 77)	2 (73)		4 weeks (33 of 77); 8 weeks (57 of 77); 16 weeks (63 of 77)	12 (73 of 77)
Soares, 2012 # [42]	Brazil (Minas Gerais)	1 (30 of 34); 2 (2 of 34); 4 (2 of 34)	n. r.		n. r.	n. r.
Andrade, 2008 # [43]	Brazil (Minas Gerais, Belo Horizonte)	n. r.	2 (20 of 20)		n. r.	12
Vasconcelos-Santos, 2009 # [44]	Brazil (Minas Gerais)	1 (10 of 11); 2 (1 of 11)	2; prednisolone where necessary		n. r.	12
Higa, 2014 # [45]	Brazil (Parana)	4 (4 of 4)	n. r.		n. r.	n. r.
Carvalheiro, 2005 # [46]	Brazil (Ribeirao Preto, Sao Paulo)	n. r.	2; 1		n. r.	12
de Melo Inagaki, 2012 # [47]	Brazil (Sergipe)	n. r.	2		n. r.	12
Gómez, 2005 # [48]	Colombia (Quindio)	1; 4 (2 of 26)	2		12 weeks; 16 weeks	0.25

Note. N = population, n = subpopulation, n. r. = not reported, written in bold = mean values of all subsamples, solid line = separation of different patient samples, dashed line = studies belong to the same overall patient pool (studies belonging to a common patient pool are also marked in light grey).

## Data Availability

Not applicable.

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
