# Peer review of "Long-Term Outcomes in Children with Congenital Toxoplasmosis—A Systematic Review"

_pathogens, 2022, doi:10.3390/pathogens11101187_

Round 1

Reviewer 1 Report

In this review, the authors present data from 31 peer-reviewed manuscripts that describe the clinical post-birth outcomes of children with congenital toxoplasmosis. The review is a great resource for data acquired in Europe, North and South America. The review of the literature emphasizes how early diagnosis and treatment of congenital toxoplasmosis leads to improved clinical outcomes for patients.

Major point:

The criteria used to exclude records (n = 149) in Figure 1 was unclear. Please describe or clarify how the records were chosen for inclusion/exclusion.

Minor point:

In Tables 1 and 2, why were some rows outlined in gray? I missed why these studies were highlighted.

Author Response

Reviewer 1:

In this review, the authors present data from 31 peer-reviewed manuscripts that describe the clinical post-birth outcomes of children with congenital toxoplasmosis. The review is a great resource for data acquired in Europe, North and South America. The review of the literature emphasizes how early diagnosis and treatment of congenital toxoplasmosis leads to improved clinical outcomes for patients.

We’d like to express our thanks to the reviewer for their critical comments supporting our manuscript.

Major point: The criteria used to exclude records (n = 149) in Figure 1 was unclear. Please describe or clarify how the records were chosen for inclusion/exclusion.

This information has been added in lines 360-365.

Minor point:In Tables 1 and 2, why were some rows outlined in gray? I missed why these studies were highlighted.

The tables have been re-designed and the coloration and separation lines been clarified in a footnote.

Reviewer 2 Report

The reviewer is delighted to thank the authors for their concise review on CT.

The authors performed a systematic review on the published data concerning the outcome of children with congenital toxoplasmosis (CT). 

Although treatment of toxoplasmosis in pregnancy and childhood virtually has not changed for decades, scientific evidence of it’s benefits is scarce. Apparently outcomes of CT differ between different regions, most probably due to different relevant strains - and possibly due to different screening- and treatment-approaches in pregnancy. 

By reviewing the relevant literature of the last decades, the authors report on data collected since the 1980 - with a sobering view on the scarcity thereof...

The conclusions consistent with the evidence and arguments

presented and they address the main question posed.

The references are appropriate.

The authors present one graph clearly explaining the methodology and papers screened. The tables represent a condensed summary of the papers included in the review. 

Author Response

The reviewer is delighted to thank the authors for their concise review on CT. The authors performed a systematic review on the published data concerning the outcome of children with congenital toxoplasmosis (CT). 

Thanks for that favourable and encouraging vote.

Although treatment of toxoplasmosis in pregnancy and childhood virtually has not changed for decades, scientific evidence of it’s benefits is scarce.

To provide a point of knowledge update based on current evidence is what drove us to undertake this meta-analysis.

Apparently outcomes of CT differ between different regions, most probably due to different relevant strains - and possibly due to different screening- and treatment-approaches in pregnancy. 

Indeed, things have become more difficult with the recognition of parasite strain impact and the geographic region where affected patients are born. We therefore tried to provide this relevant information within the table which allows to extract outcomes by geographic region.

By reviewing the relevant literature of the last decades, the authors report on data collected since the 1980 - with a sobering view on the scarcity thereof...
The conclusions consistent with the evidence and arguments presented and they address the main question posed.

We updated the tables with recent publications namely from South America to further support out conclusions.

The references are appropriate.

14 additional papers have been included in this updated version

The authors present one graph clearly explaining the methodology and papers screened. The tables represent a condensed summary of the papers included in the review. 

Thanks for a generally favourable vote.